# Reliability, Validity, and Optimal Cut-Off Scores of Action Research Arm Test and Jebsen–Taylor Hand Function Test in People with Parkinson’s Disease

**DOI:** 10.3390/healthcare13243280

**Published:** 2025-12-13

**Authors:** Sefa Eldemir, Burhanettin Cigdem

**Affiliations:** 1Faculty of Health Sciences, Department of Physiotherapy and Rehabilitation, Sivas Cumhuriyet University, Sivas 58140, Türkiye; 2Department of Neurology, School of Medicine, Sivas Cumhuriyet University, Sivas 58140, Türkiye; drbcigdem@gmail.com

**Keywords:** Jebsen–Taylor Hand Function Test, Action Research Arm Test, Parkinson’s disease, hand function, reliability, validity

## Abstract

**Highlights:**

**What are the main findings?**

The Jebsen–Taylor Hand Function Test demonstrated excellent test–retest reliability, concurrent validity, and discriminant validity, with derived cut-off scores (ranging from 3.56 to 64.23) that effectively discriminated between patients with Parkinson’s disease and healthy controls.The Action Research Arm Test exhibited excellent test–retest reliability and concurrent validity but demonstrated insufficient sensitivity and specificity for identifying patients with mild Parkinson’s disease.

**What are the implications of the main findings?**

The Jebsen–Taylor Hand Function Test is a reliable and valid assessment tool for assessing manual dexterity in patients with Parkinson’s disease.The Action Research Arm Test is a reliable assessment tool for assessing manual dexterity in patients with Parkinson’s disease.

**Abstract:**

**Background/Objectives**: Although upper extremity dexterity problems are frequently reported in people with Parkinson’s disease (PwPD), valid and reliable scales for assessing upper extremity function and dexterity are limited. The objective of this study was to investigate the reliability and validity of the Action Research Arm Test (ARAT) and the Jebsen–Taylor Hand Function Test (JTHFT) in PwPD. **Methods**: Seventy PwPD and thirty HC were recruited. The test–retest reliability was evaluated by determining the intraclass correlation coefficient (ICC). MDC_95_ was calculated by using ICC results. The concurrent validities of JTHFT and ARAT were determined by investigating their relationship with the Nine-Hole Peg Test (9-HPT), Hoehn and Yahr scale (H & Y), Unified Parkinson’s Disease Rating Scale (UPDRS), and motor symptoms (UPDRS-III). The cut-off times that best discriminated between PwPD and HC were investigated by plotting receiver operating characteristic (ROC) curves. **Results**: The ARAT and JTHFT showed excellent test–retest reliability (ICC = 0.937 to 0.995). The MDC_95_ values for the ARAT were 0.38 for the dominant hand and 0.58 for the non-dominant hand. MDC_95_ values for the JTHFT subtests and total scores ranged from 0.38 to 4.71. The ARAT, JTHFT subtests, and total scores demonstrated a fair-to-strong correlation with other outcomes (*p* < 0.05). The cut-off times that best differentiated JTHFT subtests and total scores ranged from 3.56 to 64.23. **Conclusions**: The JTHFT is a reliable and valid measurement tool for the assessment of manual dexterity in PwPD, while the ARAT is a reliable assessment tool in PwPD but does not have discriminant validity.

## 1. Introduction

Parkinson’s disease (PD) ranks as the second most prevalent neurodegenerative disorder among older adults [1]. People with Parkinson’s disease (PwPD) typically exhibit hallmark motor symptoms, including bradykinesia, involuntary dyskinesia, rigidity, tremors at rest, and fluctuating motor control (“on-off” episodes) [1]. These motor symptoms impair the performance of activities of daily living (ADLs) at different stages of the disease, such as reaching, grasping, dressing, and feeding, due to the deterioration of fine motor control [2]. Upper extremity function problems are frequently reported in PwPD as well, especially manual dexterity dysfunction, which is a common symptom of PD [3].

Accurate evaluation of upper extremity function and manual dexterity is critical for assessing treatment efficacy and disease progression in PD. The Jebsen–Taylor Hand Function Test (JTHFT) and Action Research Arm Test (ARAT) assess the ability to move the arm and manual dexterity [4,5]. Developed by Jebsen et al., the JTHFT is a standardized tool for assessing hand function and dexterity essential for ADLs, including writing, page turning, picking up small objects, feeding, stacking checkers, picking up large light objects, and picking up large heavy objects [6]. The test is recognized for different neurological diseases, including stroke, cerebral palsy, and PD. Berardi et al. showed that JTHFT was reliable and valid for stroke patients in the Italian population [7]. Similarly, Tofani et al. determined that the test had excellent internal consistency and was a valid assessment tool when used in children with cerebral palsy aged 6–18 years in the Italian population [8]. Previous studies also investigated the reliability and validity of JTHFT for PD [4,9]. These studies demonstrated that JTHFT is a reliable and easily administered assessment tool for assessing the upper extremity functions and manual dexterity of PwPD. However, the minimum detectable change scores and clinically meaningful cut-off scores distinguishing PwPD from healthy controls remain undetermined for JTHFT in PwPD. The ARAT evaluates upper extremity function and manual dexterity on both sides, such as grasp, grip, pinch, and gross movement [10]. The ARAT is a performance-based measure that examines functional upper extremity use through object manipulation tasks, providing a structured qualitative assessment of dexterity, thus reflecting real-world arm activity limitations [11]. The ARAT was developed in stroke patients by Lyle [10] and is considered by many in the field of stroke rehabilitation to be one of the most comprehensive quantitative measures of motor impairment following cortical lesion [12]. The ARAT has previously been shown to have excellent inter-rater and test–retest reliability for stroke patients [11]. It has also been investigated for evaluating upper extremity and manual dexterity impairment in people with multiple sclerosis (PwMS) and PwPD [13,14]. Carpinella et al. showed that the ARAT was a valid tool for evaluation of upper extremity motor function by examining its relationship with the 9-hole peg test (9-HPT) in MS [13]. Song evaluated the reliability of the ARAT in PwPD, demonstrating excellent test–retest reliability [14]. The study findings support the use of the ARAT as a reliable functional assessment tool in both clinical and research settings for PwPD. But the test has not been validated for PwPD. Furthermore, the minimum detectable change scores and clinically meaningful cut-off scores distinguishing PwPD from healthy controls remain undetermined in the literature.

The objective of this study was to investigate the reliability and validity of the JTHFT and ARAT in PwPD and determine the minimum detectable change scores and the cut-off scores that best discriminate PwPD from healthy people.

## 2. Materials and Methods

### 2.1. Study Design and Setting

This cross-sectional study was approved by the Health Sciences Research Ethics Committee before the recruitment of the first participants and conducted in accordance with the Declaration of Helsinki at the Department of Neurology, Sivas Cumhuriyet University.

### 2.2. Participants

According to the Consensus-Based Standards for Selection of Health Status Measurement Instruments (COSMIN) checklist, the number of individuals required for adequate internal consistency and construct validity is recommended to be 50–99 [15]. Therefore, we aimed to include at least 50 PwPD in our study. The inclusion criteria for PwPD were as follows: a diagnosis of PD according to the UK Brain Bank criteria, had Hoehn and Yahr (H & Y) stages I–III, was aged 40–80 years, and had a Mini-Mental State Examination score ≥24. PwPD were excluded if they had additional neurological disorders (e.g., stroke, visual problems, and sensory disorders of the upper extremity), disabling dyskinesia, or any musculoskeletal disorder and/or surgeries in which upper extremity evaluation is not appropriate. Additionally, individuals with the following conditions were also excluded: functional or drug-induced movement disorders, prominent apraxia, a history of alcohol or substance abuse, the use of medications that can significantly affect fine motor control or cognitive function (e.g., benzodiazepines, sedatives), systemic or psychiatric conditions that may influence speed or dexterity (e.g., major depression, hypothyroidism). Moreover, health controls (HC) (aged 40–80), recruited from the community through poster advertisements and subjected to the same exclusion criteria as PwPD, were included in the study to compare and determine JTHFT and ARAT cut-off values. Written informed consent was obtained from all participants in the study.

### 2.3. Procedures

The demographic variables and clinical characteristics of PwPD were recorded. The validity of the JTHFT and ARAT was investigated using other assessment tools. All PwPD performed the Nine-Hole Peg Test (9-HPT), JTHFT, and ARAT. To examine the intra-rater reliability of the ARAT and JTHFT, the rater reassessed 30 out of the 70 participants the next week. Healthy controls (HC) were matched based on age, sex, and BMI, and their demographic variables were recorded. HC performed the JTHFT and ARAT on one day, and the test times were compared with PwPD and used to determine the cut-off time that best discriminates between PwPD and HC. Care was taken to assess each PwPD in the ON state approximately 60 min after receiving Levodopa.

### 2.4. Outcomes Measures

The 9-HPT is a standardized assessment of manual dexterity, measuring performance speed and coordination [16]. The test setup includes nine cylindrical pegs (32 mm length, 7 mm diameter) and a board with corresponding holes (7.5 mm diameter and 13 mm depth). Participants are instructed to pick up each peg individually, insert it into a hole as rapidly as possible, and then remove the pegs one by one using the same hand [16]. In this study, the 9-HPT was performed with both dominant and non-dominant hands, and completion times (in seconds) were recorded separately for each hand.

The JTHFT is a standardized tool for assessing hand function and dexterity essential for ADLs [6]. The test consists of seven tasks: sentence writing, turning cards, picking up small objects, stimulating feeding, stacking checkers, picking up large light objects, and picking up large heavy objects [9]. In this study, the JTHFT was performed with both dominant and non-dominant hands, and all subtest times were recorded, as well as the total score separately for each hand. A lower score indicates better dexterity; a higher score indicates worse dexterity.

The ARAT is a performance-based test that examines functional upper extremity use through object manipulation tasks, consisting of a wooden box, which is placed on a table in front of the patient, containing blocks and objects of different sizes [10,13]. The ARAT consists of four subtests: grasp, grip, pinch, and gross movement. The grasp, grip, and pinch tests examine the ability to grasp, transport, and release objects of varying size, weight, and shape, with specific movement requirements. The grip subtest includes two items requiring composite movements: horizontal displacement combined with either (a) vertical motion and pronation (water pouring task) or (b) supination (washer rotation task). The pinch subtest includes six items assessing precision grip by requiring participants to pick up marbles of different sizes using thumb–digit combinations (index, middle, and ring fingers) and transfer them to a designated holder. The gross movement subtest evaluates three fundamental patterns: hand-to-mouth movement, hand-to-head placement, and hand-behind-head placement. Performance is scored on a 4-point ordinal scale (0 = no movement; 1 = partial movement; 2 = abnormal completion; 3 = normal execution). The highest score that can be obtained from the test is 57, and higher scores mean that the individual has better manual dexterity.

The Unified Parkinson’s Disease Rating Scale (UPDRS) is one of the most commonly used to monitor PD-related disability and impairment [17,18]. The scale has four components: Part I (4 items), Mentation, Behavior, and Mood; Part II (13 items), Activities of Daily Living; Part III (14 items), Motor; and Part IV (11 items), Complications. Parts I-III are scored between 0 and 4 ratings (0 = normal, 1 = slight, 2 = mild, 3 = moderate, and 4 = severe); Part IV is scored with yes and no. In this study, Part II (UPDRS-II) and III (UPDRS-III) scores were used in validity analyses.

### 2.5. Statistics

The SPSS version 23 software (SPSS, Chicago, IL, USA) was used for all statistical analyses. Shapiro–Wilk was used to test for the normality of data distribution. The demographic variables and clinical characteristics were presented as mean ± standard deviation (SD) for scale values and frequency for nominal or ordinal values. The outcome measures are presented as mean ± SD or median (interquartile range, IQR) according to the normality of data. The independent samples t-test or the Mann–Whitney U-test was used for comparisons between groups, depending on whether the data were normally distributed.

The test–retest reliabilities of the ARAT and JTHFT were investigated using the intraclass correlation coefficient (ICC) based on a two-way random effects analysis of variance and a 95% confidence interval (CI) [19]. Test–retest reliability was categorized as poor-to-moderate if the ICC was <0.75, good if it ranged between 0.75 and 0.89, and excellent if it was ≥0.90 [20]. The standard error of measurement (SEM) and minimal detectable change (MDC_95_) were calculated by using ICC results according to the following formulas:SEM = SD × √(1 − ICC)MDC_95_ = 1.96 × SEM × √2

Concurrent and discriminant validity were investigated in this study. For concurrent validity, the relationship between JTHFT and ARAT and other outcome measurements (9-HPT, H & Y, UPDRS-III, and UPDRS-total) was examined. Pearson’s (r) or Spearman’s (ρ) correlation coefficients were used depending on whether the data were normally distributed. The correlation coefficients were categorized as excellent (≥0.9), strong (0.80 to 0.89), moderate (0.51 to 0.75), fair (0.26 to 0.50), or poor (0 to 0.25) [20]. For discriminant validity, the JTHFT-subtests, JTHFT-total, and ARAT-total times were compared between PwPD and HC and evaluated using t-tests or Mann–Whitney U-tests on whether the data were normally distributed. Additionally, the cut-off times that best discriminated between PwPD and HC were investigated by plotting receiver operating characteristic (ROC) curves. The best cut-off times were determined by using Youden´s index.

## 3. Results

This study included 70 PwPD (41 men, 29 women; mean age, 62.90 ± 9.25) and 30 HC (18 men, 12 women; mean age, 59.53 ± 11.18). Table 1 presents the demographic variables and clinical characteristics of participants.

Table 2 presents the ICC, SEM, MDC_95_, and cut-off values of the ARAT and JTHFT. The ARAT and JTHFT showed excellent test–retest reliability (ICC values ranging from 0.937 to 0.995). The MDC_95_ values for the subtests of the ARAT were as follows: dominant and non-dominant hands were 0.38 and 0.58, respectively. The MDC_95_ values for the subtests of the JTHFT were as follows: dominant and non-dominant hands for the Writing sentences subtest were 2.58 and 3.99, respectively; dominant and non-dominant hands for the simulated page turning subtest were 2.27 and 3.74, respectively; dominant and non-dominant hands for the lifting small objects subtest were 2.23 and 1.41, respectively; dominant and non-dominant hands for the stacking checkers subtest were 1.88 and 1.14, respectively; dominant and non-dominant hands for the simulated feeding subtest were 2.19 and 2.77, respectively; dominant and non-dominant hands for the lifting large light objects subtest were 0.61 and 0.94, respectively; dominant and non-dominant hands for the lifting large heavy objects subtest were 0.55 and 0.69, respectively, and dominant and non-dominant hands for the JTHFT-total were 4.71 and 4.35, respectively. The study encountered a significant ceiling effect in ARAT measurement scores, where a significant proportion of PwPD, particularly within the [positive/negative] group, achieved maximum (Dominant hand; 72.9% (51 participants) and nondominant hand; 74.03% (52 participants)) or near-maximum scores. Clustering of data at the upper end of the scale resulted in insufficient variability to meaningfully distinguish between groups. As a result, the discriminatory capacity of the measure was compromised, preventing the generation of a reliable receiver operating characteristic (ROC) curve. Therefore, the cut-off value could not be calculated for the ARAT test. The optimum JTHFT subtests cut-off times to distinguish between PwPD and HC were determined to be 14.97 (writing sentences with dominant hand), 28.69 (writing sentences with non-dominant hand), 4.73 (simulated page turning with dominant hand), 4.75 (simulated page turning with non-dominant hand), 6.96 (lifting small objects with dominant hand), 7.78 (lifting small objects-with non-dominant hand), 3.56 (stacking checkers with dominant hand), 3.59 (stacking checkers with non-dominant hand), 8.32 (simulated feeding with dominant hand), 10.14 (simulated feeding with non-dominant hand), 3.60 (lifting large light objects with dominant hand), 3.82 (lifting large light objects with non-dominant hand), 3.97 (lifting large heavy objects with dominant hand), 3.93 (lifting large heavy objects with non-dominant hand), 48.70 (JTHFT-total with dominant hand), and 64.23 (JTHFT-total with non-dominant hand) seconds.

Table 3 presents Pearson’s or Spearman’s correlation coefficients between the ARAT and JTHFT and the 9-HPT, H & Y, UPDRS-III, and UPDRS-total measures. The ARAT scores demonstrated fair-to-moderate negative correlation with JTHFT subtest (except non-dominant side writing sentences and simulated feeding, ρ = −0.21–ρ = −0.46, *p* ≤ 0.040) 9-HPT (ρ = −0.56–ρ = −0.58, *p* < 0.001), H & Y (ρ = −0.36–ρ = −0.40, *p* < 0.001), UPDRS-III (ρ = −0.48–ρ = −0.50, *p* < 0.001), and UPDRS-total (ρ = −0.43–ρ = −0.45, *p* < 0.001). Unlike other tests, the ARAT test, with higher scores indicating better manual dexterity, showed a negative correlation. This suggests that the ARAT measurements correlated well with other scales. The JTHFT subtest scores, with increasing scores indicating better performance, demonstrated fair-to-strong-positive correlation with 9-HPT (ρ = 0.80–ρ = 0.37, *p* ≤ 0.006), H & Y (ρ = 0.62–ρ = 0.27, *p* ≤ 0.022), UPDRS-III (ρ = 0.58–ρ = 0.33, *p* ≤ 0.006), and UPDRS-total (r = 0.60–ρ = 0.25, *p* ≤ 0.034).

Table 4 presents a comparison between PwPD and HC. Significant differences were in all ARAT and JTHFT subtest scores between the groups (*p* ≤ 0.002).

## 4. Discussion

This study investigated the test–retest reliability, SEM, MDC_95_, concurrent validity, discriminant validity, and cut-off value for the ARAT, JTHFT subtests, and JTHFT-total scores in PwPD. The findings demonstrate that the JTHFT subtests and JTHFT-total scores are valid and reliable measures in PwPD. In addition, the study determined cut-off scores for the JTHFT that best differentiate PwPD from HC. The ARAT had excellent test–retest reliability and concurrent validity but lacked sufficient sensitivity and specificity for identifying mild PwPD. Therefore, no clinically useful cut-off value could be determined.

ARAT, which was first developed for people with stroke, is preferred in many neurological diseases, especially stroke [10,13,14]. Chen et al. demonstrated that the ARAT has moderate-to-excellent predictive validity (ρ = 0.58–0.66) and high reliability (person reliability = 0.94) and high internal consistency (Cronbach’s alpha = 0.97) in assessing upper extremity dysfunction in stroke patients [11]. In addition, Nijland et al. demonstrated that ARAT has moderate-to-excellent inter-rater and intra-rater reliability (ICC = 0.70–0.97 and ICC: 0.92–0.97, respectively) and moderate-to-strong concurrent validity (r = 0.70–0.86) in people with stroke [21]. Similarly, Sharif et al. demonstrated that ARAT has moderate-to-excellent intra-rater reliability (ICC = 0.935–0.995) and fair concurrent validity (r = 0.11) in people with stroke [22]. Carpinella et al. examined the concurrent validity of ARAT in PwMS by looking at its correlation with 9-HPT. According to the study results, 50% of mild PwMS received full scores from ARAT, while a moderate negative correlation (r = −0.776–−0.765) was found in mild-to-moderate PwMS [13]. Song et al. demonstrated high test–retest reliability (ICC: 0.93–0.99) of the ARAT in PwPD with H & Y stages II–III [14]. To the researchers’ knowledge, no previous study has been conducted to evaluate the SEM, MDC_95_, cut-off values, and discriminant validity of the ARAT in PwPD. Our study findings showed excellent test–retest reliability and fair-to-moderate concurrent validity, consistent with previous studies, although disease groups differed except for one study. Additionally, we found a significant difference when compared with HC for discriminant validity. However, we found that individuals with PwPD did not reach sufficient value for the cut-off value investigated with the ROC curve and therefore did not have sufficient sensitivity and specificity. Carpinella et al. reported in their study that, like our study, half of mild PwMS received full scores, which may have created a ceiling effect [13]. They also stated that individuals with mild involvement showed the lowest deviation from normative data and that the scale may be inadequate in assessing the status of these patients. In our study, individuals with mild-to-moderate involvement (H & Y: I–III) were evaluated, and it was found that a high percentage of individuals received maximum scores. This situation was found to cause a ceiling effect in the ARAT and to compromise measurement precision. Based on this, we recommend that disease stages be taken into consideration when investigating the validity of the ARAT test in future studies.

The JTHFT is a comprehensive tool that evaluates hand function and dexterity required for activities of daily living, such as writing, card turning, feeding simulation, and picking up and carrying objects of different sizes [6]. Berardi et al. found high internal consistency (Cronbach’s alpha; 0.96 for dominant hand and 0.92 for non-dominant) and fair-to-moderate concurrent validity (r = 0.68–0.34) for the subtests of the JTHFT in Italian stroke patients [7]. Ferreiro et al. demonstrated that JTHFT has good-to-excellent inter-rater (ICC = 1.0) and intra-rater reliabilities (ICC = 0.997–0.999) and good internal consistency (Cronbach’s alpha = 0.924) in people with stroke [23]. Sığırtmaç et al. demonstrated that JTHFT has good-to-excellent test–retest reliability (ICC = 0.84–0.97) and fair concurrent validity (r = −0.33–0.39) in people with hand injuries. Additionally, they found that the cut-off value of the total score was found to be 37.08 s for the injured hand [24]. Tofani et al. determined that the test had excellent internal consistency (Cronbach’s alpha = 0.944 and 0.911) and moderate-to-good concurrent validity (r = 0.53–0.74) [8]. The validity and reliability of the JTHFT in many different disease groups have been demonstrated in the previous studies mentioned above. There are two studies investigating the psychometric properties of the JTHFT in PwPD. Mak et al. found that the JTHFT subtests (ICC = 0.39–0.94) and total scores (ICC = 0.89–0.97) had poor-to-excellent test–retest reliability in PwPD [9]. Additionally, individuals were found to complete all JTHFT subtests and total scores, except dominant side writing, in significantly longer times than healthy individuals. In another study, Galeoto et al. found that the subtests and total score of the JTHFT had acceptable internal consistency (Cronbach’s alpha = 0.556–0.668) and moderate-to-excellent test–retest reliability (ICC = 0.754–0.988) [4]. The study examined the correlation with the dynamometer for concurrent validity and found that it showed moderate-to-strong correlations with many JTHFT subtests (r = 0.54–0.88). In addition, no correlation was found between H & Y, motor fluctuations, dyskinesia, and the JTHFT in the study (*p* > 0.05). Consistent with previous studies, the JTHFT subtests and total scores showed excellent test–retest reliability (ICC = 0.937–0.995) and concurrent validity (fair-to-strong correlation with 9-HPT and ARAT (ρ = 0.80–ρ = −0.21)) in our study. In the study, concurrent validity was also supported by examining the correlations between the ARAT and JTHFT subtest and total scores with disease severity (H & Y), motor symptoms (UPDRS-III), and disease-related disability and impairment (UPDRS-total). Moreover, SEM, MDC_95_, discriminant validity, and cut-off values for JTHFT subtests and total scores were determined for the first time. The findings of this study provided detailed information regarding the validity and reliability of the ARAT and JTHFT in PwPD.

The findings of the present study hold significant implications for both clinical practice and future research. The findings of this study demonstrate that the JTHFT is a reliable and valid measurement tool for assessing manual dexterity in PwPD. At the same time, the ARAT is a reliable assessment tool. This study provides reliable and standardized tools to precisely measure upper extremity dexterity. This can directly contribute to the development of more standardized assessment protocols in Parkinson’s disease clinics, ensuring that dexterity deficits are identified and monitored consistently over time. Furthermore, the high reliability and validity of these tests make them excellent outcome measures for guiding and personalizing rehabilitation strategies. For example, specific subtests of the JTHFT can help therapists determine which functional tasks (e.g., feeding, writing) are most impaired, thus enabling the design of highly targeted, task-specific training programs. Calculated MDC values are particularly important for clinical decision making as they provide a concrete threshold for determining whether a change in a patient’s performance after intervention is real and clinically meaningful beyond measurement error. Future research should focus on applying these tools in intervention studies to monitor disease progression and to robustly evaluate the effectiveness of new rehabilitation techniques on manual dexterity.

This study had several limitations. First, the sample included individuals with mild-to-moderate PD stages. Therefore, the findings of this study cannot be generalized to all PwPD. Secondly, in this study, patients were not separated according to disease severity or gender. This was because the sample size was not large enough to allow reliable analyses across subgroups. We recommend that validity and reliability studies be conducted in different stages of PwPD and gender in future studies. We believe that moderate and advanced stage PwPD can provide more reliable information, especially for the validity of the ARAT. Thirdly, responsiveness was not investigated in this study. The addition of this test in further studies will guide us in better understanding the effects of the scales. Finally, inter-rater and intra-rater reliability were not investigated in this study. Conducting these analyses in subsequent studies may provide evidence for the multidimensional examination of reliability.

## 5. Conclusions

All the ARAT, JTHFT subtests, and JTHFT-total scores demonstrated excellent test–retest reliability in evaluating manual dexterity in PwPD. The MDC_95_ values of the ARAT, JTHFT subtests, and JTHFT-total, ranging from 0.38-to-4.71, may be useful for clinicians to detect a true change in manual dexterity after an upper extremity treatment protocol. The ARAT, JTHFT subtests, and JTHFT-total scales showed fair-to-strong correlations with 9-HPT, H & Y stages, UPDRS-III, and UPDRS-total, demonstrating good concurrent validity. ARAT and JTHFT scores differed significantly between PwPD and HC. The JTHFT, with cut-off times ranging from 3.56 to 64.23 s, was found to discriminate PwPD well from HC. However, the ARAT does not have sufficient discriminant validity. In conclusion, JTHFT is a reliable and valid measurement tool for the assessment of manual dexterity in individuals with disabilities. In addition, the ARAT test is a reliable assessment tool in PwPD but does not have discriminant validity. Further studies are needed to investigate the validity of the ARAT test in detail.

## Figures and Tables

**Table 1 healthcare-13-03280-t001:** Demographic characteristics of participants.

	People with PD (n = 70)	Healthy Controls (n = 30)	*p*
Age (years)	62.90 ± 9.25	59.53 ± 11.18	0.12
BMI (kg/m^2^)	27.90 ± 5.21	27.81 ± 3.48	0.91
Gender, Female/Male (Female%)	29/41 (41.4%)	12/18 (40%)	0.89
Dominant side, right/left (right%)	70/0 (100%)	29/1(97%)	0.30
Disease duration (months)	64.48 ± 60.72	-	
Side where symptoms begin, dominant/non-dominant (dominant%)	35/34 (50%)	-	
Hoehn & Yahr stage, n (%) 1 2 3	17 (24.3%) 26 (37.1%) 27 (38.6%)	- - -	

BMI: Body mass index, PD: Parkinson’s disease, values are mean ± SD (standard deviation), median (interquartile range-IQR), or as otherwise indicated, *p* < 0.05.

**Table 2 healthcare-13-03280-t002:** Test–retest reliability, SEM, MDC_95_, and cut-off values of the Action Research Arm Test and Jebsen–Taylor Hand Function Test in people with Parkinson’s disease.

		ICC (95% CI)	MDC_95_	SEM	Cut-Off Value
ARAT (0–56)	D	0.993 (0.985–0.997)	0.38	0.14	-
	ND	0.986 (0.971–0.993)	0.58	0.21	-
JTHFT (s) Writing sentences	D	0.995 (0.990–0.998)	2.58	0.93	14.97
	ND	0.990 (0.980–0.995)	3.99	1.44	28.69
Simulated page turning	D	0.942 (0.877–0.972)	2.27	0.82	4.73
	ND	0.957 (0.909–0.979)	3.74	1.35	4.75
Lifting small objects	D	0.946 (0.886–0.974)	2.23	0.84	6.96
	ND	0.976 (0.949–0.988)	1.41	0.51	7.78
Stacking checkers	D	0.963 (0.923–0.982)	1.88	0.68	3.56
	ND	0.975 (0.947–0.988)	1.14	0.41	3.59
Simulated feeding	D	0.937 (0.868–0.970)	2.19	0.79	8.32
	ND	0.980 (0.958–0.991)	1.77	0.64	10.14
Lifting large light objects	D	0.987 (0.973–0.994)	0.61	0.22	3.60
	ND	0.951 (0.897–0.977)	0.94	0.34	3.82
Lifting large heavy objects	D	0.981 (0.961–0.991)	0.55	0.20	3.97
	ND	0.975 (0.947–0.988)	0.69	0.25	3.93
JTHFT-total	D	0.994 (0.988–0.997)	4.71	1.70	48.70
	ND	0.995 (0.990–0.998)	4.35	1.57	64.23

ARAT: Action Research Arm Test. D: dominant side, ICC: intraclass correlation coefficients, JTHFT: Jebsen–Taylor Hand Function Test, MDC_95_: minimal detectable change, ND: non-dominant side, SEM: standard error of measurement, s: seconds.

**Table 3 healthcare-13-03280-t003:** Mean values of the outcome measure correlations with the Action Research Arm Test and Jebsen–Taylor Hand Function Test in people with Parkinson’s disease.

		ARAT (0–56) D	ARAT (0–56) ND	9-HPT (s) D	9-HPT (s) ND	H & Y (1–5)	UPDRS-III (0–108)	UPDRS-Total (0–199)
ARAT (0–56)	D	- -	- -	ρ = −0.58 *p* < 0.001 *	ρ = −0.56 *p* < 0.001 *	ρ = −0.40 *p* = 0.001 *	ρ = −0.50 *p* < 0.001 *	ρ = −0.45 *p* < 0.001 *
	ND	- -	- -	ρ = −0.57 *p* < 0.001 *	ρ = −0.57 *p* < 0.001 *	ρ = −0.36 *p* = 0.002 *	ρ = −0.48 *p* < 0.001 *	ρ = −0.43 *p* < 0.001 *
JTHFT (s) Writing sentences	D	ρ = −0.31 *p* = 0.011 *	ρ = −0.26 *p* = 0.031 *	ρ = 0.47 *p* < 0.001 *	ρ = 0.39 *p* = 0.001 *	ρ = 0.38 *p* = 0.001 *	ρ = 0.43 *p* < 0.001 *	ρ = −0.44 *p* < 0.001 *
	ND	ρ = −0.21 *p* = 0.77 *	ρ = −0.14 *p* = 0.219	ρ = 0.37 *p* = 0.007 *	ρ = 0.41 *p* < 0.001 *	ρ = 0.27 *p* = 0.022 *	ρ = 0.21 *p* = 0.086	ρ = 0.26 *p* = 0.033 *
Simulated page turning	D	ρ = −0.40 *p* = 0.001 *	ρ = −0.36 *p* = 0.002 *	ρ = 0.63 *p* < 0.001 *	ρ = 0.45 *p* < 0.001 *	ρ = 0.35 *p* = 0.003 *	ρ = 0.33 *p* = 0.006 *	ρ = 0.25 *p* = 0.034 *
	ND	ρ = −0.44 *p* < 0.001 *	ρ = −0.46 *p* < 0.001 *	ρ = 0.70 *p* < 0.001 *	ρ = 0.77 *p* < 0.001 *	ρ = 0.58 *p* < 0.001 *	ρ = 0.58 *p* < 0.001 *	ρ = 0.51 *p* < 0.001 *
Lifting small objects	D	ρ = −0.39 *p* = 0.001 *	ρ = −0.33 *p* = 0.005 *	ρ = 0.72 *p* < 0.001 *	ρ = 0.49 *p* < 0.001 *	ρ = 0.49 *p* < 0.001 *	ρ = 0.50 *p* < 0.001 *	ρ = 0.45 *p* < 0.001 *
	ND	ρ = 0.34 *p* = 0.004 *	ρ = −0.34 *p* = 0.005 *	ρ = 0.59 *p* < 0.001 *	ρ = 0.62 *p* < 0.001 *	ρ = 0.62 *p* < 0.001 *	r = 0.58 *p* < 0.001 *	r = 0.60 *p* < 0.001 *
Stacking checkers	D	ρ = 0.34 *p* = 0.004 *	ρ = −0.26 *p* = 0.031 *	ρ = 0.61 *p* < 0.001 *	ρ = 0.53 *p* < 0.001 *	ρ = 0.50 *p* < 0.001 *	ρ = 0.53 *p* < 0.001 *	ρ = 0.51 *p* < 0.001 *
	ND	ρ =0.35 *p* = 0.003 *	ρ = −0.31 *p* = 0.010 *	ρ = 0.58 *p* < 0.001 *	ρ = 0.69 *p* < 0.001 *	ρ = 0.53 *p* < 0.001 *	ρ = 0.52 *p* < 0.001 *	ρ = 0.50 *p* < 0.001 *
Simulated feeding	D	ρ = −0.36 *p* = 0.002 *	ρ = −0.25 *p* = 0.040 *	ρ = 0.37 *p* = 0.002 *	ρ = 0.37 *p* = 0.006 *	ρ = 0.40 *p* < 0.001 *	ρ = 0.41 *p* < 0.001 *	ρ = 0.38 *p* = 0.001 *
	ND	ρ = −0.02 *p* = 0.857	ρ = −0.04 *p* = 0.728	ρ = 0.26 *p* = 0.056	ρ = 0.47 *p* = 0.002 *	ρ = 0.42 *p* < 0.001 *	ρ = 0.33 *p* = 0.005 *	ρ = 0.35 *p* = 0.003 *
Lifting large light objects	D	ρ = −0.36 *p* = 0.002 *	ρ = −0.28 *p* = 0.018 *	ρ = 0.62 *p* < 0.001 *	ρ = 0.56 *p* < 0.001 *	ρ = 0.42 *p* < 0.001 *	ρ = 0.41 *p* < 0.001 *	ρ = 0.40 *p* = 0.001 *
	ND	ρ = −0.35 *p* = 0.003 *	ρ = −0.29 *p* = 0.015 *	ρ = 0.57 *p* < 0.001 *	ρ = 0.72 *p* < 0.001 *	ρ = 0.52 *p* < 0.001 *	r = 0.41 *p* < 0.001 *	r =0.40 *p* = 0.001 *
Lifting large heavy objects	D	ρ = −0.29 *p* = 0.014 *	ρ = −0.22 *p* = 0.071 *	ρ = 0.59 *p* < 0.001 *	ρ = 0.54 *p* < 0.001 *	ρ = 0.43 *p* < 0.001 *	r = 0.40 *p* = 0.001 *	r = 0.41 *p* < 0.001 *
	ND	ρ = −0.44 *p* < 0.001 *	ρ = −0.38 *p* = 0.001 *	ρ = 0.67 *p* < 0.001 *	ρ = 0.80 *p* < 0.001 *	ρ = 0.53 *p* < 0.001 *	r = 0.47 *p* < 0.001 *	r = 0.45 *p* < 0.001 *
JTHFT-total	D	ρ = −0.43 *p* < 0.001 *	ρ = −0.37 *p* = 0.001 *	ρ = 0.71 *p* < 0.001 *	ρ = 0.61 *p* < 0.001 *	ρ = 0.59 *p* < 0.001 *	ρ = 0.58 *p* < 0.001 *	ρ = 0.55 *p* < 0.001 *
	ND	ρ = −0.38 *p* = 0.001 *	ρ = −0.31 *p* =0.009 *	ρ = 0.58 *p* < 0.001 *	ρ = 0.75 *p* < 0.001 *	ρ = 0.57 *p* < 0.001 *	ρ = 0.47 *p* < 0.001 *	ρ = 0.48 *p* < 0.001 *

9-HPT: Nine-Hole Peg Test, ARAT: Action Research Arm Test, D: dominant side, H & Y: Hoehn and Yahr, JTHFT: Jebsen–Taylor Hand Function Test, ND: non-dominant side, r: Pearson’s correlation coefficient, s: seconds, UPDRS: Unified Parkinson’s Disease Rating Scale, UPDRS-II: UPDRS-activity of daily living, UPDRS-III: UPDRS-motor score, ρ: Spearman´s correlation coefficient. * *p* < 0.05.

**Table 4 healthcare-13-03280-t004:** Discriminant validity of the Action Research Arm Test and Jebsen–Taylor Hand Function Test.

		People with PD (n = 70)	Healthy Controls (n = 30)	*p* ^a^
ARAT (0–56)	D	57 (56–57)	57 (57–57)	0.002
	ND	57 (56–57)	57 (57–57)	0.002
JTHFT (s) Writing sentences	D	17.06 (13.95–29.58)	13.08 (9.89–15.65)	<0.001
	ND	39.07 (34.54–48.65)	23.63 (16.59–27.79)	<0.001
Simulated page turning	D	6.41 (5.29–8.29)	4.19 (3.56–4.67)	<0.001
	ND	6.76 (5.05–8.87)	3.89 (3.36–4.67)	<0.001
Lifting small objects	D	9.24 (7.12–11.26)	5.81 (5.34–6.93)	<0.001
	ND	9.49 (7.50–12.01)	6.31 (5.39–7.82)	<0.001
Stacking checkers	D	4.23 (3.58–5.31)	2.91 (2.45–3.55)	<0.001
	ND	4.66 (3.66–6.11)	3.23 (2.70–3.55)	<0.001
Simulated feeding	D	10.08 (8.90–11.62)	7.33 (6.66–7.59)	<0.001
	ND	12.06 (10.77–15.20)	7.73 (6.69–9.64)	<0.001
Lifting large light objects	D	4.76 (3.72–6.00)	3.09 (2.63–3.60)	<0.001
	ND	4.88 (3.92–6.22)	3.32 (2.82–3.72)	<0.001
Lifting large heavy objects	D	5.00 (4.09–5.81)	3.42 (2.99–3.95)	<0.001
	ND	4.77 (4.25–6.03)	3.29 (2.95–3.67)	<0.001
JTHFT-total	D	61.20 (50.92–75.43)	39.63 (34.84–47.12)	<0.001
	ND	85.20(73.15–100.49)	51.02 (47.42–55.51)	<0.001

ARAT: Action Research Arm Test, D: dominant side, JTHFT: Jebsen–Taylor Hand Function Test, ND: non-dominant side, s: seconds, values are median (interquartile range—IQR). *p* < 0.05. ^a^ Mann–Whitney U test.

## Data Availability

The raw data supporting the conclusions of this article have been stored on the corresponding author’s personal computer and will be made available by the authors upon request.

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
