# Peer review of "Reliability, Validity, and Optimal Cut-Off Scores of Action Research Arm Test and Jebsen–Taylor Hand Function Test in People with Parkinson’s Disease"

_healthcare, 2025, doi:10.3390/healthcare13243280_

Round 1
Reviewer 1 Report
Comments and Suggestions for Authors
This is a well-executed psychometric validation study examining two widely used upper-extremity function tests – the Action Research Arm Test (ARAT) and the Jebsen Taylor Hand Function Test (JTHFT) – in people with Parkinson’s disease (PwPD). The authors carefully evaluate test-retest reliability, concurrent validity, discriminant validity, and minimal detectable change (MDC), also determining cut-off scores to distinguish PwPD from healthy controls (HC).
The topic is clinically relevant and methodologically robust as dexterity impairment is a major functional concern in Parkinson’s disease, and validated upper-limb measures are limited. The manuscript is clearly structured and well written and acknowledges the limitations of ARAT, such as the ceiling effect. The analyses are rigorous, and the findings have direct application in neurorehabilitation and clinical assessment. A few minor revisions would enhance clarity and translational impact.
Major Comments:
- The correlations with UPDRS and 9-HPT are well documented. However, the authors should interpret directionality more clearly: higher JTHFT times reflect worse dexterity, while higher ARAT scores reflect better function. This inverse relationship could be confusing to readers unfamiliar with both scales.
- The JTHFT cut-off times are informative, but it would be valuable to explain how these thresholds can guide clinical decision-making, such as differentiating between mild and moderate functional impairment or tracking therapy response.
- The discussion is comprehensive but somewhat descriptive. It could be improved by briefly addressing practical implications, including but not limited to how clinicians can integrate JTHFT results in PD rehabilitation planning or how MDC95 can guide individualized goal setting.
Minor Comments:
- The authors should clarify if testing was performed while patients were on medication to ensure motor-state consistency.
- Did the authors mean “responsiveness” on line 335?
- Some sentences, such as “The Action Research Arm Test had …. specificity for mild in patients with Parkinson's disease” on line 19-21, sound grammatically incorrect.
- The authors should proofread the document and check for any grammatical errors.
Author Response
Comment 1: [This is a well-executed psychometric validation study examining two widely used upper-extremity function tests – the Action Research Arm Test (ARAT) and the Jebsen Taylor Hand Function Test (JTHFT) – in people with Parkinson’s disease (PwPD). The authors carefully evaluate test-retest reliability, concurrent validity, discriminant validity, and minimal detectable change (MDC), also determining cut-off scores to distinguish PwPD from healthy controls (HC).
The topic is clinically relevant and methodologically robust as dexterity impairment is a major functional concern in Parkinson’s disease, and validated upper-limb measures are limited. The manuscript is clearly structured and well written and acknowledges the limitations of ARAT, such as the ceiling effect. The analyses are rigorous, and the findings have direct application in neurorehabilitation and clinical assessment. A few minor revisions would enhance clarity and translational impact.]
Response 1: We would like to thank the reviewer for the positive comments. We've included our responses to each suggestion below.
Comment 2: [Major Comments:
The correlations with UPDRS and 9-HPT are well documented. However, the authors should interpret directionality more clearly: higher JTHFT times reflect worse dexterity, while higher ARAT scores reflect better function. This inverse relationship could be confusing to readers unfamiliar with both scales.]
Response 2: Thanks for the suggestion. To avoid confusion, we have added “A lower score indicates better dexterity; a higher score indicates worse dexterity.” for JTHFT and “The highest score that can be obtained from the test is 57, and higher scores mean that the individual has better manual dexterity.” for ARAT in the Outcomes measures section. We also revised the third paragraph of the Results section to make it clearer:
“Table 3 presents Pearson’s or Spearman’s correlation coefficients between the ARAT and JTHFT and the 9-HPT, H&Y, UPDRS-III, and UPDRS-Total measures. The ARAT test scores demonstrated fair to moderate negative correlation with JTHFT subtest (except non- dominant side Writing sentences and Simulated feeding, ρ= -0.21- ρ= -0.46, p≤0.040) 9-HPT (ρ= -0.56- ρ= -0.58, p<0.001), H&Y (ρ= -0.36- ρ= -0.40, p<0.001), UPDRS-III (ρ= -0.48- ρ= -0.50, p<0.001), and UPDRS-Total (ρ= -0.43- ρ= -0.45, p<0.001). Unlike other tests, the ARAT test, with higher scores indicating better manual dexterity, showed a negative cor-relation. This suggests that the ARAT measurements correlated well with other scales. The JTHFT subtest scores, with increasing scores indicating better performance, demonstrated fair to strong positive correlation with 9-HPT (ρ=0.80- ρ=0.37, p≤0.006), H&Y (ρ=0.62- ρ=0.27, p≤0.022), UPDRS-III (ρ=0.58- ρ=0.33, p≤0.006), and UPDRS-Total (r=0.60- ρ=0.25, p≤0.034).”
Comment 3: [The JTHFT cut-off times are informative, but it would be valuable to explain how these thresholds can guide clinical decision-making, such as differentiating between mild and moderate functional impairment or tracking therapy response.]
Response 3: Thank you again for your suggestion for subgroup analysis. We carefully considered this valuable suggestion. The COSMIN guidelines, which set methodological standards for studies assessing measurement properties, recommend a minimum of 50 participants for test-retest reliability and construct validity analyses to obtain reliable and valid results. The overall sample size of our current study (n=70) meets this recommendation. However, performing analyses by dividing the data into groups may affect the statistical reliability of the subgroup analysis results. Therefore, to maintain the methodological soundness of our study, we deemed it appropriate not to present subgroup analysis in the manuscript. Furthermore, we addressed this in the limitations section, which is the final paragraph of the Discussion section. The relevant section in the limitations is as follows: "...Secondly, in this study, patients were not separated according to disease severity or gender. This was because the sample size was not large enough to allow reliable analyses across subgroups. We recommend that validity and reliability studies be conducted in different stages of PwPD and gender in future studies... "
Comment 4: [The discussion is comprehensive but somewhat descriptive. It could be improved by briefly addressing practical implications, including but not limited to how clinicians can integrate JTHFT results in PD rehabilitation planning or how MDC95 can guide individualized goal setting.]
Response 4: Thank you for your suggestion. We've added the following paragraph to the discussion section of this manuscript to better explain the benefits and future perspectives for clinicians:
"The findings of the present study hold significant implications for both clinical practice and future research. The findings of this study demonstrate that the JTHFT is a reliable and valid measurement tool for assessing manual dexterity in PwPD. At the same time, the ARAT test is a reliable assessment tool. This study provides reliable and standardized tools to precisely measure upper extremity dexterity. This can directly contribute to the development of more standardized assessment protocols in Parkinson's disease clinics, ensuring that dexterity deficits are identified and monitored consistently over time. Furthermore, the high reliability and validity of these tests make them excellent outcome measures for guiding and personalizing rehabilitation strategies. For example, specific subtests of the JTHFT can help therapists determine which functional tasks (e.g., feeding, writing) are most impaired, thus enabling the design of highly targeted, task-specific training programs. Calculated MDC values ​​are particularly important for clinical decision making as they provide a concrete threshold for determining whether a change in a patient's performance after intervention is real and clinically meaningful beyond measurement error. Future research should focus on applying these tools in intervention studies to monitor disease progression and to robustly evaluate the effectiveness of new rehabilitation techniques on manual dexterity."
Comment 5: [Minor Comments:
The authors should clarify if testing was performed while patients were on medication to ensure motor-state consistency.]
Response 5: Thank you for the suggestion. We agree with the reviewer that this information is an important detail for the assessments. Therefore, we paid attention to it throughout the study. Following the reviewer's suggestion, we have added the following sentence under the Procedures heading to clarify this information:
"Care was taken to assess each PwPD in the ON state approximately 60 minutes after receiving Levodopa."
Comment 6: [Did the authors mean “responsiveness” on line 335?]
Response 6: We apologize for the error. Following the reviewer's suggestion, we have edited the sentence as follows:
“Thirdly, responsiveness was not investigated in this study.”
Comment 7: [Some sentences, such as “The Action Research Arm Test had …. specificity for mild in patients with Parkinson's disease” on line 19-21, sound grammatically incorrect.
The authors should proofread the document and check for any grammatical errors.
Response 7: We agree with the reviewer that some sentences are not very understandable. Following the reviewer's suggestion, the Highlights heading has been revised and rewritten as follows. The language check was carried out throughout the text, and corrections have been highlighted in yellow.
“Highlights
What are the main findings?
- The Jebsen-Taylor Hand Function Test demonstrated excellent test-retest reliability, concurrent validity, and discriminant validity, with derived cut-off scores (ranging from 3.56 to 64.23) that effectively discriminated between patients with Parkinson's disease and healthy controls.
- The Action Research Arm Test exhibited excellent test-retest reliability and concur-rent validity but demonstrated insufficient sensitivity and specificity for identifying patients with mild Parkinson's disease.”
Reviewer 2 Report
Comments and Suggestions for Authors
Thank you very much for inviting me to review this interesting manuscript investigating the reliability and validity of the Action Research Arm Test (ARAT) and the Jebsen Taylor Hand Function Test (JTHFT) in people with Parkinson’s disease (PwPD). The study is well designed, addresses an important clinical issue, and provides useful insights into the quantitative assessment of upper limb dexterity in this population.
However, some aspects should be revised and clarified before considering the manuscript for publication:
-
Exclusion criteria:
Although neurological and musculoskeletal causes seem to have been excluded, the authors should clarify whether other potential confounding factors were ruled out, such as:-
Functional movement disorders;
-
The presence of apraxia;
-
Substance abuse (e.g., alcohol or other toxins);
-
The use of medications that can affect fine motor control (benzodiazepines, sedatives, etc.);
-
Systemic conditions that may influence ideomotor speed or dexterity (e.g., depression, hypothyroidism).
-
-
Subgroup analysis:
It would be valuable to perform a subgroup analysis comparing patients with mild vs. moderate Parkinson’s disease, to determine whether the reliability and discriminant validity of the ARAT and JTHFT differ according to disease severity. -
Future perspectives:
The authors should briefly discuss the future implications of their findings — for instance, how these results could contribute to developing standardized dexterity assessment protocols or guide rehabilitation strategies in PwPD.
Overall, the manuscript is promising and addresses a relevant gap in the literature, but these revisions would significantly strengthen the work and its clinical applicability.
Author Response
Comment 1: [Thank you very much for inviting me to review this interesting manuscript investigating the reliability and validity of the Action Research Arm Test (ARAT) and the Jebsen Taylor Hand Function Test (JTHFT) in people with Parkinson’s disease (PwPD). The study is well designed, addresses an important clinical issue, and provides useful insights into the quantitative assessment of upper limb dexterity in this population.]
Response 1: We would like to thank the reviewer for the positive comments. We've included our responses to each suggestion below.
Comment 2: [However, some aspects should be revised and clarified before considering the manuscript for publication:
Exclusion criteria:
Although neurological and musculoskeletal causes seem to have been excluded, the authors should clarify whether other potential confounding factors were ruled out, such as:
Functional movement disorders;
The presence of apraxia;
Substance abuse (e.g., alcohol or other toxins);
The use of medications that can affect fine motor control (benzodiazepines, sedatives, etc.);
Systemic conditions that may influence ideomotor speed or dexterity (e.g., depression, hypothyroidism).]
Response 2: Thank you for the suggestion. We obtained detailed personal information for all individuals before assessments (such as medications used, medical diagnoses, past medical history, and family history). Therefore, we have data that we can revise according to the reviewer's suggestion. To clarify the exclusion criteria, we have revised the text as follows:
“PwPD were excluded if they had additional neurological disorders (e.g., stroke, visual problems, sensory disorders of the upper extremity), disabling dyskinesia, or any musculoskeletal disorder and/or surgeries in which upper extremity evaluation is not appropriate. Additionally, individuals with the following conditions were also excluded: functional or drug-induced movement disorders, prominent apraxia, a history of alcohol or substance abuse, the use of medications that can significantly affect fine motor control or cognitive function (e.g., benzodiazepines, sedatives), systemic or psychiatric conditions that may influence speed or dexterity (e.g., major depression, hypothyroidism).”
Comment 3: [Subgroup analysis:
It would be valuable to perform a subgroup analysis comparing patients with mild vs. moderate Parkinson’s disease, to determine whether the reliability and discriminant validity of the ARAT and JTHFT differ according to disease severity.]
Response 3: Thank you again for your suggestion for subgroup analysis. We carefully considered this valuable suggestion. The COSMIN guidelines, which set methodological standards for studies assessing measurement properties, recommend a minimum of 50 participants for test-retest reliability and construct validity analyses to obtain reliable and valid results. The overall sample size of our current study (n=70) meets this recommendation. However, performing analyses by dividing the data into groups may affect the statistical reliability of the subgroup analysis results. Therefore, to maintain the methodological soundness of our study, we deemed it appropriate not to present subgroup analysis in the manuscript. Furthermore, we addressed this in the limitations section, which is the final paragraph of the Discussion section. The relevant section in the limitations is as follows: "...Secondly, in this study, patients were not separated according to disease severity or gender. This was because the sample size was not large enough to allow reliable analyses across subgroups. We recommend that validity and reliability studies be conducted in different stages of PwPD and gender in future studies..."
Comment 4: [Future perspectives:
The authors should briefly discuss the future implications of their findings — for instance, how these results could contribute to developing standardized dexterity assessment protocols or guide rehabilitation strategies in PwPD.]
Response 4: Thanks for the suggestion. We've added the following paragraph to the discussion section:
"The findings of the present study hold significant implications for both clinical practice and future research. The findings of this study demonstrate that the JTHFT is a reliable and valid measurement tool for assessing manual dexterity in PwPD. At the same time, the ARAT test is a reliable assessment tool. This study provides reliable and standardized tools to precisely measure upper extremity dexterity. This can directly contribute to the development of more standardized assessment protocols in Parkinson's disease clinics, ensuring that dexterity deficits are identified and monitored consistently over time. Furthermore, the high reliability and validity of these tests make them excellent outcome measures for guiding and personalizing rehabilitation strategies. For example, specific subtests of the JTHFT can help therapists determine which functional tasks (e.g., feeding, writing) are most impaired, thus enabling the design of highly targeted, task-specific training programs. Calculated MDC values ​​are particularly important for clinical decision making as they provide a concrete threshold for determining whether a change in a patient's performance after intervention is real and clinically meaningful beyond measurement error. Future research should focus on applying these tools in intervention studies to monitor disease progression and to robustly evaluate the effectiveness of new rehabilitation techniques on manual dexterity."
Comment 5: [Overall, the manuscript is promising and addresses a relevant gap in the literature, but these revisions would significantly strengthen the work and its clinical applicability.]
Response 5: We thank the reviewer for the suggestions. We have revised our manuscript based on the suggestions and hope that it has reached a satisfactory level in its current form.
Round 2
Reviewer 2 Report
Comments and Suggestions for Authors
Well done!